# The Impact of Prostate Volume on the Prostate Imaging and Reporting Data System (PI-RADS) in a Real-World Setting

**DOI:** 10.3390/diagnostics13162677

**Published:** 2023-08-15

**Authors:** Yannic Volz, Maria Apfelbeck, Nikolaos Pyrgidis, Paulo L. Pfitzinger, Elena Berg, Benedikt Ebner, Benazir Enzinger, Troya Ivanova, Michael Atzler, Philipp M. Kazmierczak, Dirk-André Clevert, Christian Stief, Michael Chaloupka

**Affiliations:** 1Department of Urology, LMU Klinikum, Ludwig-Maximilians University, Marchioninistr. 15, 81377 Munich, Germany; maria.apfelbeck@med.uni-muenchen.de (M.A.); nikolaos.pyrgidis@med.uni-muenchen.de (N.P.); paulo.pfitzinger@med.uni-muenchen.de (P.L.P.); elena.berg@med.uni-muenchen.de (E.B.); benedikt.ebner@med.uni-muenchen.de (B.E.); benazir.enzinger@med.uni-muenchen.de (B.E.); troya.ivanova@med.uni-muenchen.de (T.I.); michael.atzler@med.uni-muenchen.de (M.A.); christian.stief@med.uni-muenchen.de (C.S.); michael.chaloupka@med.uni-muenchen.de (M.C.); 2Interdisciplinary Ultrasound-Center, Department of Radiology, LMU Klinikum, Ludwig-Maximilians University, Marchioninistr. 15, 81377 Munich, Germany; philipp.kazmierczak@med.uni-muenchen.de (P.M.K.); dirk.clevert@med.uni-muenchen.de (D.-A.C.)

**Keywords:** prostate volume, PI-RADS, biopsy, diagnostic accuracy, prostate cancer, MRI

## Abstract

Multiparametric magnetic resonance imaging (mpMRI) has emerged as a new cornerstone in the diagnostic pathway of prostate cancer. However, mpMRI is not devoid of factors influencing its detection rate of clinically significant prostate cancer (csPCa). Amongst others, prostate volume has been demonstrated to influence the detection rates of csPCa. Particularly, increasing volume has been linked to a reduced cancer detection rate. However, information about the linkage between PI-RADS, prostate volume and detection rate is relatively sparse. Therefore, the current study aims to assess the association between prostate volume, PI-RADS score and detection rate of csP-Ca, representing daily practice and contemporary mpMRI expertise. Thus, 1039 consecutive patients with 1151 PI-RADS targets, who underwent mpMRI-guided prostate biopsy at our tertiary referral center, were included. Prior mpMRI had been assessed by a plethora of 111 radiology offices, including academic centers and private practices. mpMRI was not secondarily reviewed in house before biopsy. mpMRI-targeted biopsy was performed by a small group of a total of ten urologists, who had performed at least 100 previous biopsies. Using ROC analysis, we defined cut-off values of prostate volume for each PI-RADS score, where the detection rate drops significantly. For PI-RADS 4 lesions, we found a volume > 61.5 ccm significantly reduced the cancer detection rate (OR 0.24; 95% CI 0.16–0.38; *p* < 0.001). For PI-RADS 5 lesions, we found a volume > 51.5 ccm to significantly reduce the cancer detection rate (OR 0.39; 95% CI 0.25–0.62; *p* < 0.001). For PI-RADS 3 lesions, none of the evaluated clinical parameters had a significant impact on the detection rate of csPCa. In conclusion, we show that enlarged prostate volume represents a major limitation in the daily practice of mpMRI-targeted biopsy. This study is the first to define exact cut-off values of prostate volume to significantly impair the validity of PI-RADS assessed in a real-world setting. Therefore, the results of mpMRI-targeted biopsy should be interpreted carefully, especially in patients with prostate volumes above our defined thresholds.

## 1. Introduction

Prostate cancer (PCa) remains the most frequent cancer in men worldwide [1]. Over the course of the last few years, commonly used tools for the detection of prostate cancer, such as the serum prostate-specific antigen (PSA), digital rectal examination (DRE) and prostate biopsy, have been supplemented with multiparameter magnetic resonance imaging (mpMRI) as well as mpMRI-guided biopsy of the prostate (Fbx) [2,3]. The diagnostic accuracy of these tools has been demonstrated in the literature [2,3,4]. Based on the previous notion, current guidelines recommend using mpMRI as a diagnostic tool in the detection of PCa [5]. The Prostate Imaging Reporting and Data System (PI-RADS) is the structured reporting system for the assessment of mpMRI of the prostate [6]. Beyond its potential to enable MRI-guided biopsy of the prostate, studies also showed the potential of mpMRI to predict adverse pathology at the time of radical prostatectomy [7]. However, it has been postulated that several factors may influence the PI-RADS scoring system in the detection of clinically significant prostate cancer (csPCas). In particular, the experience of the radiologist assessing the mpMRI [8] and the experience of the physician performing the biopsy may impact the detection rate of csPCa [9,10]. Yet, the literature remains sparse about the effect of clinical parameters on the detection rate of csPCa after Fbx and its possible influence on the validity of the PI-RADS scoring system. Previous studies showed an association between prostate volume and index lesion size for patients receiving in-bore MRI-guided biopsies [11] as well as in cognitive fusion biopsies [12]. In general, it has been shown that the detection rate of csPCa may decrease with increasing prostate volume, especially in glandular volumes above 40 ccm [13,14]. Furthermore, prostate volume itself may impact the incidence and aggressiveness of PCa [15]. Still, no studies have explored the association between prostate volume and the detection rate of a csPCa in relation to the PI-RADS score. In the outpatient clinic of our department, FBx was performed by multiple urologists according to an mpMRI assessed by a heterogenous group of 111 radiology offices. Therefore, the aim of the current study was to assess the impact of the prostate volume on the detection rate of csPCa by FBx, representing a real-world setting.

## 2. Materials and Methods

### 2.1. Population and Data Collection

We retrospectively analyzed a prospectively maintained database of all MRI-guided transrectal prostate biopsies performed at our tertiary referral center between March 2015 and August 2022. The present study was approved by the local ethics committee (#22-0318), and its findings are reported based on the Strengthening the Reporting of Observational Studies in Epidemiology (STROBE) statement for cohort studies [16]. Patients with suspicious findings on mpMRI (PI-RADS 3, 4 or 5) were referred to our tertiary referral center by either their urologist or by the outpatient clinic of our department. The mpMRI was either performed at the Department of Radiology of Ludwig-Maximilian University of Munich or by a heterogeneous group of radiology offices, including other hospitals, private practices and radiology offices in foreign countries. MpMRI was not reviewed by a local domestic radiologist at our department before MRI-guided biopsy. csPCa was defined as a Gleason score above/equal to 3 + 4 = 7a, as used in previous studies [17]. 

The following clinical parameters were recorded: age at FBx, serum level of PSA, prostate volume, PSA density (calculated by PSA divided by prostate volume (ng/mL/cm^3^)), history of previous prostate biopsies, findings of the DRE and assessment of the index lesion of mpMRI according to the PI-RADS. Prostate volume was calculated as recommended in current PI-RADS^®^ v2.1 [18,19] by using the ellipsoid formula (i.e., maximum anterior–posterior dimension × maximum longitudinal dimension × maximum transverse dimension × 0.52) in most of the cases. Yet, there are also manual or automated segmentation methods for prostate volume calculation. Anterior–posterior and longitudinal diameters were measured on the mid-sagittal T2-weighted images and the transverse diameter on the axial T2-weighted images, respectively. Furthermore, histopathological data included the number of positive biopsy cores, grading of PCa according to the International Society of Uropathologists (ISUP) and the ratio of tumor infiltration per PCa positive biopsy core. The histopathological examination was performed at the Department of Pathology at LMU Klinikum in Munich, Germany. 

### 2.2. MRI-Guided Biopsy

Fbx was performed by a group of ten experienced urologists with at least 100 Fbx performed independently. Each biopsy included the index lesions with at least 3 cores taken from each lesion as well as an additional systematic randomized biopsy with 6 cores from the left and right lobe, respectively (base, mid and apical gland) according to current guidelines [20]. The fusion of the mpMRI and real-time ultrasound was performed using plane wise fusion. The axial T2-weighted MRI sequence was used for image fusion. Software and Hardware for assessment of FBx was provided by Epiq7, Philips Percunav, Philips Medical Systems, Bothell, WA, USA. 

### 2.3. Statistical Analysis

A per-patient and a per-target analysis was performed. All continuous variables were assessed for normality, were summarized as mean with standard deviation (SD) and the corresponding comparisons were performed with the two-sample *t*-test or the analysis of variance. All categorical variables were summarized as absolute numbers with proportions and were compared with the chi-squared (χ2) test. To identify an optimal cut-off for prostate volume that predicts the results of the biopsy in different PI-RADS targets, we used the Youden’s index based on receiver operating characteristic (ROC) analysis. In particular, the Youden’s index was calculated for each point of the ROC curve for prostate volume and biopsy results in PI-RADS 3, 4 and 5 targets, and its maximum value was selected as a criterion for estimating the optimal cut-off point for prostate volume. Subsequently, a univariate logistic regression analysis was undertaken for PI-RADS 3, 4 and 5 targets to evaluate the role of this prostate volume cut-off in predicting the results of the biopsy. These findings were also adjusted for age and initial PSA through multivariable logistic regression. For all analyses, odds ratios (ORs) and 95% confidence intervals (CIs) were estimated. The statistical calculations were performed in the R statistical software (Version 3.6.3), and two-sided *p*-values lower than 0.05 were considered statistically significant. 

## 3. Results

### 3.1. Population Characteristics and Histopathological Findings

Baseline patient characteristics are displayed in Table 1. 

A total of 1039 patients and a total of 1151 mpMRI targets were included. Patients had a mean age of 67.6 ± 8.2 years and a mean prostate volume of 55.9 ± 34.1 cm^3^. Overall, the mpMRI reports included 203/1151 (17.6%) PI-RADS 3 targets, 560/1151 (48.7%) PI-RADS 4 targets and 388/1151 (33.7%) PI-RADS 5 targets. 

The detection rate of csPCa was 394/1039 (37.9%) in patient-based analysis. In the target-based analysis, the detection rate of csPCa was 21/203 (10.4%) in PI-RADS 3 lesions, 183/560 (32.7%) in PI-RADS 4 lesions and 228/388 (58.8%) in PI-RADS 5 lesions, respectively. Further, 89/249 (35.7%) patients had a positive DRE but negative biopsy result. Prostate volume was significantly smaller in patients with a csPCa-positive target biopsy compared to csPCa-negative target biopsies (61.3 ccm vs. 47.1 ccm; *p* < 0.001) in per-patient analysis. As seen in Figure 1, patients with detection of csPCa in PI-RADS 5 lesions had a significantly lower prostate volume compared to patients without detection of csPCa in PI-RADS 5 lesions (*p* = 0.023) (Table 2).

### 3.2. Impairment of PI-RADS by Prostate Volume 

By calculating the maximum Youden’s index based on the ROC analysis (Figure 2), the following prostate volume cut-off values of impairment of PI-RADS classification were evaluated: For PI-RADS 3 targets, the cut-off value was a volume ≥ 43.5 ccm (not significant).For PI-RADS 4, a prostate volume of ≥61.5 ccm.For PI-RADS 5, a volume ≥ 51.5 ccm.

As shown in Table 3, in the multivariable analysis adjusting for age and iPSA, all patients with a PI-RADS 4 target lesion and a prostate volume ≥ 61.5 ccm undergoing FBx showed a significant impact on the csPCa detection rate (OR 0.24 95% CI 0.16–0.38; *p* ≤ 0.001) compared to a volume < 61.5 ccm. Similarly, in the case of a PI-RADS 5 target lesion, a volume above the cut-off value of ≥51.5 ccm showed a significant impact on the detection of csPCa (OR 0.39; 95% CI 0.25–0.62; *p* < 0.001). Nevertheless, age and iPSA were not identified as predictors for positive target biopsies (*p* = 0.088 and *p* = 0.1, respectively). For PI-RADS 3 lesions, prostate volume, age and iPSA did not significantly impact csPCa detection rates (*p* = 0.11, *p* = 0.4 and *p* = 0.6, respectively). 

Regarding PI-RADS 4 lesions, patients with a prostate volume ≥ 61.5 ccm showed a significantly lower csPCa detection rate of 35/199 (18%) compared to patients with a prostate volume < 61.5 ccm 148/361 (41%) (*p* < 0.001). Regarding PI-RADS 5 lesions, patients with a prostate volume ≥ 51.5 ccm showed a significantly lower csPCa detection rate of 68/143 (48%) compared to patients with a prostate volume < 51.5 ccm 160/245 (65%) (*p* < 0.001). 

## 4. Discussion

Since PCa is the most frequently occurring cancer in males, ensuring diagnostic efficiency while safely detecting clinically significant prostate cancer (csPCa) remains a challenge in order to avoid over- or under-treatment. The use of mpMRI and the PI-RADS score helped to improve csPCa detection rates [3,21]; however, they are not devoid of factors influencing the detection rates. Various factors, such as the radiologist’s experience, the lesion location or the version of PI-RADS used, have been shown to impact the detection of csPCa [8,10,22]. However, the impact of commonly observed clinical parameters, such as prostate volume, is not yet fully established. The current study, therefore, aimed to assess the impact of prostate volume on the detection rate of csPCa for each PI-RADS score.

We performed mpMRI-guided biopsies on a large number of patients at our tertiary referral center. MpMRIs were performed at our radiological department, private radiological practices or other hospitals. Therefore, our cohort represents a real-world analysis.

Between March 2015 and August 2022, we included a total number of 1039 biopsies in the current study. The overall detection rate for csPCa was 37.9%, which is comparable to detection rates reported in previous studies [12,23,24]. Similarly, the detection rates for each PI-RADS score also matched previously published csPCa detection rates for the respective PI-RADS score [11,12,25]. 

Patients with a negative biopsy of the mpMRI target had significantly larger prostates compared to men with a positive mpMRI target biopsy (61.3 ccm vs. 47.1 ccm, *p* < 0.001). This finding confirms results of a systematic review by Knight et al. [26], summarizing 12 studies that described an inverse relationship between prostate volume and PCa incidence. The authors postulated that a larger prostate may be protective of PCa or impact the detection rates of csPCa itself. Similarly, Nepal et al. were able to show a decreased detection rate for csPCa when there was an increase in prostate volume above 40 ccm [13]. Yet, they did not distinguish between different PI-RADS values. Therefore, these results underline the need to assess cut-off values where a certain prostate volume may lead to a decreased detection rate of csPCa.

Prior research demonstrated that the detection rates for mpMRI target lesions are influenced by their location. The transitional zone, in particular, continues to pose a diagnostic challenge. This is likely due to the benign presence of prostatic hyperplasia in nearly all elderly patients and causing interfering radiological features in mpMRI [27,28,29]. Additionally, it has been shown that BPH nodules as well as prostatitis can mimic the radiological features of PCa [30,31,32]. Another study demonstrated that even with diffusion-weighted imaging, the accuracy of detecting a csPCa was significantly higher (82.2% vs. 67.1%; *p* = 0.002) in the peripheral zone than in the transitional zone [33]. Another explanation could be the hypothesis that the peripheral zone is being compressed by the expanding transitional zone in the case of benign prostatic hyperplasia. This finding was made in a recently published study by Lin et al., where the authors reported significantly reduced glandular tissue volume of the peripheral zone in larger prostates [34].

Consequently, the aim of the current study was to establish cut-off values for each PI-RADS score, with the intention of improving the diagnostic accuracy and interpretation of mpMRI results. For PI-RADS 3, we evaluated a cut-off value of 43.5 ccm. However, in the multivariable analysis, the cut-off value did not impact the cancer detection significantly. This may be due to the overall ambivalent detection rate of PI-RADS 3 that has been well documented in the literature [35,36]. The detection rate of csPCa for PI-RADS 3 was only 21/432 (10.4%) in our cohort and, therefore, the influence of factors, such as age or volume, might be underpowered in these cases. On the contrary, for PI-RADS 4, we evaluated that a prostate volume above 61 ccm leads to significantly lower detection rates of csPCa (48% > 61.5 vs. 65% < 61.5 ccm, *p* < 0.001). Similarly, increasing age and PSA levels were associated with a higher risk of csPCa detection (OR 1.03 95% CI 1.00–1.05; *p* = 0.035 and OR 1.04 95% CI 1.01–1.06; *p* = 0.002, respectively).

Interestingly, for PI-RADS 5, we evaluated a prostate volume cut-off value of 51.5 ccm. Compared to age and PSA, prostate volume was the only significant factor that significantly influenced the detection rates of csPCa (volume: OR 0.48; 95% CI 0.25–0.62; *p* < 0.001; age: OR 1.02 95% CI 1.00–1.05; *p* = 0.088; iPSA: OR 1.02; 95% CI 1.00–1.04; *p* = 0.100). This may be mainly due to the fact that a PIRADS 5 lesion results in csPCa diagnosis in most cases [2]. The overall detection rate for csPCa in PI-RADS 5 lesions in our study was 228/388 (58.8%). The detection rate of csPCa in PI-RADS 5 lesions in prostates <51.5 ccm was 160/245 (65%) compared to 68/143 (48%) (*p* < 0.001). Based on our findings, increasing the prostate volume above 51.5 ccm seems to negatively affect csPCa detection and, therefore, urologists should consider this impairment when planning FBx in large prostates. Our findings are in line with the study of Elkhoury et al., in which the csPCa detection rate was lower with increasing prostate volume for all biopsy methods [37]. They also found decreasing PCa detection rates with increasing prostate volume. The detection rates were 32/42 (77.0%) for small volumes, 98/156 (62.8%) for moderate volumes and 21/50 (42.0%) for high volumes and also decreased statistically significantly (*p* = 0.006) [37]. However, to our knowledge, this is the first study to determine a certain threshold for csPCa based on the PI-RADS in a heterogenous and interdisciplinary real-world setting. Even though the cut-off values of 61.5 ccm for PI-RADS 4 lesions and 51.5 ccm for PI-RADS 5 lesions may not apply in external cohorts without validation yet, they offer the practical benefit of representing a landmark, where the results of MRI-guided biopsy should be interpreted with caution. 

In an effort to overcome this challenge of MRI-guided biopsy, upscaling of the number of biopsy cores could potentially improve the detection rate of csPCa in larger prostates. In general, the optimal number of biopsy cores is still controversially discussed. While some studies suggest a PI-RADS-adjusted approach, others show that even two cores can detect the vast majority of csPCa [38,39]. In a large retrospective evaluation of 451 patients undergoing transrectal MRI-guided biopsy, Beetz et al. showed that the detection rate of csPCa was not improved by adding a fourth or fifth MRI-guided biopsy. In fact, the most relevant histopathology was diagnosed by the first three MRI-targeted biopsy cores in 97% of patients [40]. This is consistent with our study protocol, where three biopsy cores were taken independently of the PI-RADS score of the index lesion. So far, no study has evaluated the detection rate of csPCa by varying the numbers of biopsy cores dependent on the prostate volume.

It should be noted that there are several limitations mitigating the findings of the present study. First and foremost, it is a single-center retrospective analysis of a prospectively maintained database with a potential inherent selection bias. Also, missing data might cause substantial bias. Still, 111 different radiology offices and ten urologists performing FBx were involved, reflecting real-world data. Experience in performing MRI-guided biopsy could potentially affect the detection rate of csPCa in our study. However, all involved urologists performed a minimum number of 100 biopsies per year. In a retrospective analysis of 377 patients, Bevill et al. showed that MRI-guided biopsy is associated with a learning curve of approximately 100 cases. The authors suggested that four or five biopsy cores should be taken during the initial learning curve, but three cores per index lesion are sufficient thereafter [41]. We defined cut-off values of prostate volume that can help urologists interpret the results of the biopsy. However, our data lack external validation. Thus, prospective studies are needed to evaluate and strengthen these cut-off values. Our findings were further mitigated by the lack of a central secondary radiology review for the PIRADS score, as well as by the different levels of experience of the radiologists evaluating the mpMRI and the urologists performing the Fbx. It should also be stressed that the lack of data on certain baseline characteristics (e.g., DRE status, previous biopsy) did not permit us to add more risk factors in the multivariable analysis. Furthermore, we did not include the location of the index lesion, which may also impact the cancer detection rate. Another limitation is the different method of calculation of the prostate volume, since more than just the ellipsoid formula was used according to the PI-RADS v2.1. However, we believe that all different methods of prostate volume calculation represent the actual prostate volume within its known measurement variations. Yet, different methods of calculation could lead to potential bias and, therefore, the results should be interpreted carefully. Finally, we could not determine whether the extraction of more biopsy cores in larger prostates would have diminished the observed differences in the detection rates of csPCa.

## 5. Conclusions

MpMRI-guided biopsy of the prostate remains one of the most important tools in the diagnostic pathway of prostate cancer. However, several factors may influence the cancer detection rate and, thus, lead to incorrect treatment decisions. To our knowledge, our study is the first to identify cut-off values of prostate volume impairing PI-RADS accuracy when performing mpMRI-guided biopsy of the prostate in a real-world setting. Patients with prostate volumes above these cut-off values may display decreased detection rates of csPCa cancer. Therefore, urologists should interpret biopsy results with caution in patients with large prostate volumes.

## Figures and Tables

**Figure 1 diagnostics-13-02677-f001:**
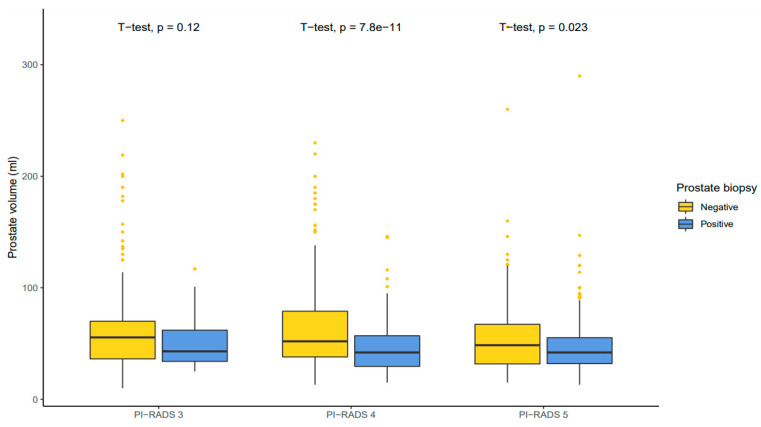
Positive/negative biopsies (cancer detection rate) and PI-RAD score.

**Figure 2 diagnostics-13-02677-f002:**
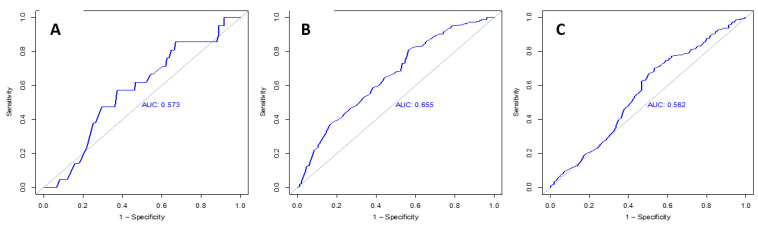
The ROC curves with the AUC on which the Youden’s index was based. (**A**) for PI-RADS 3, (**B**) for PI-RADS 4 and (**C**) for PI-RADS 5. AUC: are under the curve; ROC: receiver operating characteristic.

**Table 1 diagnostics-13-02677-t001:** Baseline characteristics of the included patients. Continuous values are presented as mean and standard deviation (±); categorical values are given as number (*n*; %). Documentation of clinical data is incomplete throughout the presented cohort regarding history of prior biopsy in 965/1039 patients (92.2%) and DRE in 686/1039 (66%). iPSA: initial prostate-specific antigen; DRE: digital rectal examination.

Characteristic	Overall, *n* = 1039	Negative Target Biopsy, *n* = 645	Positive Target Biopsy, *n* = 394	*p*-Value
Age (years)	67.6 ± 8.2	67.1 ± 7.9	68.5 ± 8.6	0.006
iPSA (ng/mL)	9.8 ± 7.8	9.3 ± 6.9	10.7 ± 8.9	0.008
PSA density (ng/mL^2^)	0.2 ± 0.1	0.2 ± 0.1	0.3 ± 0.2	<0.001
Prior biopsy (*n*)	321 (33.3%)	229 (37.7%)	92 (25.8%)	<0.001
Positive DRE	249 (36.3%)	89 (20.6%)	160 (63.0%)	<0.001
Prostate volume (mL)	55.9 ± 34.1	61.3 ± 37.1	47.1 ± 26.1	<0.001

**Table 2 diagnostics-13-02677-t002:** Target-based analysis: Gleason grade groups in relation to PI-RADS values, relation of positive biopsy cores and infiltration depth in percent.

Characteristic	Overall, *n* = 432	PI-RADS 3, *n* = 21	PI-RADS 4, *n* = 183	PI-RADS 5, *n* = 228	*p*-Value
Positive biopsy cores (%)	0.5 ± 0.2	0.4 ± 0.2	0.4 ± 0.2	0.5 ± 0.2	
Gleason Grade group					0.006
Gleason Grade group 2	180 (41.7%)	13 (61.9%)	88 (48.1%)	79 (34.6%)	
Gleason Grade group 3	66 (15.3%)	1 (4.8%)	30 (16.4%)	35 (15.4%)	
Gleason Grade group 4	143 (33.1%)	6 (28.6%)	57 (31.1%)	80 (35.1%)	
Gleason Grade group 5	43 (9.9%)	1 (4.8%)	7 (4.3%)	34 (14.9%)	
Infiltration at biopsy (%)	55.6 ± 21.3	47.0 ± 20.2	50.7 ± 19.6	60.4 ± 21.7	

**Table 3 diagnostics-13-02677-t003:** Univariate and multivariable logistic regression models for positive prostate biopsy using the cut-off for prostate volume. CI: confidence interval; OR: odds ratio.

Characteristic	Univariate			Multivariable		
	OR	95% CI	*p*-Value	OR	95% CI	*p*-Value
PIRADS 3						
Prostate volume > 43.5 ccm	0.45	0.17, 1.11	0.085	0.45	0.16, 1.18	0.11
Age	0.97	0.91, 1.02	0.2	0.97	0.92, 1.03	0.4
iPSA	1.00	0.92, 1.07	>0.9	1.02	0.94, 1.10	0.6
PIRADS 4						
Prostate volume > 61.5 ccm	0.30	0.20, 0.45	<0.001	0.24	0.16, 0.38	<0.001
Age	1.02	1.00, 1.04	0.091	1.03	1.00, 1.05	0.035
iPSA	1.02	1.00, 1.05	0.029	1.04	1.01, 1.06	0.002
PIRADS 5						
Prostate volume > 51.5 ccm	0.48	0.32, 0.73	<0.001	0.39	0.25, 0.62	<0.001
Age	1.01	0.99, 1.04	0.3	1.02	1.00, 1.05	0.088
iPSA	1.01	0.99, 1.03	0.4	1.02	1.00, 1.04	0.10

## Data Availability

The data presented in this study are available on request from the corresponding author. The data are not publicly available due to ethical and legal reasons.

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
