# Peer review of "The Impact of Prostate Volume on the Prostate Imaging and Reporting Data System (PI-RADS) in a Real-World Setting"

_diagnostics, 2023, doi:10.3390/diagnostics13162677_

Round 1

Reviewer 1 Report

This study is the first to determine cut off values of prostate volume impairing PI RAD S accuracy when performing mpMRI guided biopsy of the prostate in a real world setting. 287 Patients with prostate volumes above these cut off values may display decreased detection rates of csPCa cancer. Therefore, urologists should interpret biopsy results w ith caution in patients with large prostates volume in relation to our cut off values and PI RADS lesion. I am supportive of publication.

Author Response

This study is the first to determine cut off values of prostate volume impairing PI RAD S accuracy when performing mpMRI guided biopsy of the prostate in a real world setting. 287 Patients with prostate volumes above these cut off values may display decreased detection rates of csPCa cancer. Therefore, urologists should interpret biopsy results w ith caution in patients with large prostates volume in relation to our cut off values and PI RADS lesion. I am supportive of publication.

We thank the reviewer for taking the time to review our manuscript and for supporting our publication.

Reviewer 2 Report

The paper describes a methodology for assessing the prostate volume with regard to the detection of csPCA.

It is well documented, and provides statistically results with cut-offs for Pi-RADS 4 and 5. Results are well sustained by the methdology.

In the abstract, the sentence

"mpMRI-targeted biopsy was assessed by a group of ten urologists."

is not clear. It gets clarifies within the paper, but a better explanation would make the abstract more accurate.

There are several split words, probably from a copy-paste from the original manuscript: e.g. im-pact, repre-sents. Should be taken care on the editorial review.

Author Response

The paper describes a methodology for assessing the prostate volume with regard to the detection of csPCA.

It is well documented, and provides statistically results with cut-offs for Pi-RADS 4 and 5. Results are well sustained by the methdology.

First of all, we would like to thank the reviewer for taking the time to review our manuscript and we are happy to change the comments made by the reviewer.

In the abstract, the sentence

"mpMRI-targeted biopsy was assessed by a group of ten urologists."

is not clear. It gets clarifies within the paper, but a better explanation would make the abstract more accurate.

Again, thank you very much for this important remark. We have changed the wording accordingly and hope that is now more clear that 10 trained urologist performed the biopsies.